# Lipid Profile, Eating Habit, and Physical Activity in Children with Down Syndrome: A Prospective Study

**DOI:** 10.3390/diseases12040068

**Published:** 2024-03-29

**Authors:** Luca Pecoraro, Melissa Zadra, Francesco Cavallin, Silvana Lauriola, Giorgio Piacentini, Angelo Pietrobelli

**Affiliations:** 1Pediatric Unit, Department of Surgical Sciences, Dentistry, Gynecology and Pediatrics, University of Verona, 37126 Verona, Italy; 2Department of Medicine, University of Verona, 37134 Verona, Italy; 3Independent Researcher, 36020 Solagna, Italy; 4Pennington Biomedical Research Center, Baton Rouge, LA 70808, USA

**Keywords:** down syndrome, childhood obesity, Mediterranean diet, KIDMED questionnaire, Godin questionnaire, dyslipidemia

## Abstract

Children with Down Syndrome (DS) frequently undergo health challenges, including a higher prevalence of overweight and obesity. We aimed to evaluate the impact of dietary and physical advice provided by a specialized pediatrician over two years. In this prospective study, 44 children with DS, aged 2 to 17, underwent outpatient follow-up visits every six months between December 2020 and May 2023. Dietary habits, physical activities, anthropometric data, and laboratory results were recorded at baseline and 2-year follow-up. Adherence to the Mediterranean diet and physical activity were investigated using the ‘KIDMED’ and ‘Godin–Shepard Leisure-Time’ questionnaires, respectively, completed by the parents of the children. Venous blood samples were taken to determine the lipid profile. A significant reduction in BMI z-scores (*p* = 0.006) and an improvement in Godin–Shepard questionnaire scores (*p* = 0.0004) were observed. On the other hand, the lipid profile worsened, with an increase in LDL-c (*p* = 0.04) and a decrease in HDL-c (*p* = 0.03). Children with DS may benefit from an educational program on nutrition and physical activity to optimize weight control. Different interventions should target the lipid profile. Preventive intervention and follow-up by the pediatrician are essential for DS, which should continue into adulthood.

## 1. Introduction

Down Syndrome (DS) is the most common chromosomal disorder among live-born infants, caused by the trisomy of a part or the whole chromosome 21 [1]. This syndrome presents as a complete free trisomy, translocation, and mosaicism in 95%, 3%, and 2% of cases, respectively [2]. DS is characterized by great phenotypic variability among affected individuals, ranging from mild to severe. It is the most frequent cause of cognitive disability, which is associated with specific dysmorphic features, congenital malformations, and psychomotor developmental disorders. Its comorbidities are represented by heart defects, Alzheimer’s disease, leukemia, hypertension, and gastrointestinal diseases [1]. The life expectancy of the DS population has improved in recent years due to increased attention to treatable medical disorders in DS [3,4]. However, challenges persist regarding the health conditions of individuals with DS. In addition to the most known clinical features of DS, such as short stature and reduced head circumference, individuals with DS also face issues of overweight and obesity [5]. The prevalence of overweight and obesity in the youth DS population is higher than in the general population, with a prevalence of 23–70% [5]. Some features of DS increase susceptibility to weight gain, and these determinants include high leptin levels, low resting energy expenditure, and comorbidities such as hypothyroidism and heart defects [5,6]. These factors contribute to lower energy consumption and thus a requirement of 500–800 calories per day less than in the unaffected population; this increases the risk of excessive calorie intake and consequent weight gain. Additionally, parents may overestimate the amount of energy consumed and could overfeed their children [7]. Children with DS tend to develop inappropriate food habits due to clinical features such as hypotonia, reduced oral space, slow swallowing reflex, and altered food perception based on consistency, taste, temperature, and smell [8]. Hypotonia also affects the strength and mobility of mouth muscles, contributing to difficulties with chewing, swallowing, lip closure, and tongue protrusion and can induce gastroesophageal reflux. The combination of reduced oral space and macroglossia further leads to breathing and chewing problems, nasal passages narrowing, and respiratory secretions. The swallowing phase is slowed down by difficulty in bolus formation, caused by the failure of the jaw to close after food entry, the accumulation of food in the anterior and lateral grooves, and reduced tongue movements. These factors result in coordination issues between the chewing, swallowing, and breathing stages. Difficulties in the use of cutlery are evident, which promote the consumption of food in liquid form. Frequent episodes of vomiting after meals are also common. Additionally, there are pathologies of the gastrointestinal tract that contribute to the nutritional complexities of individuals with DS, including dental defects, malocclusion, esophageal atresia with tracheoesophageal fistula, duodenal stenosis, annular pancreas, rectal atresia, Hirschsprung’s disease, and celiac disease [9]. This leads to special food choices with preferences for simple carbohydrates that are easier to swallow and a minor intake of fruits and vegetables [10]. Moreover, children with DS tend to exceed the recommended daily amounts of protein, carbohydrates, and lipids, thus increasing their energy and macronutrient intake. In contrast, their intake of unsaturated omega-3 and omega-6 fatty acids and vitamins and minerals such as A, B9, B12, E, zinc, magnesium, iodine, and selenium is often insufficient [11]. Ensuring proper nutrition for individuals with DS is crucial yet challenging. In a recent cross-sectional study, Bialek et al. showed that most subjects, including adults and children with DS, are usually adequately fed although parents often make dietary errors [12]. General medical advice regarding good health principles is using the Mediterranean diet [13]. It is identified as a comprehensive dietary pattern, a set of eating habits that is part of a local tradition that differs between countries and mainly concerns Mediterranean localities. It refers not to a specific diet or individual foods or nutrients but is considered a lifestyle and cultural pattern. Moreover, it is considered a sustainable food model that respects the environment, biodiversity, local cultural heritages, social interaction, and economic aspects. It was, therefore, recognized as a UNESCO Intangible Cultural Heritage of Humanity in 2010 [14]. The Mediterranean diet represents a model of a healthy diet, and some components (such as fruits and vegetables, legumes, whole grains, dietary fiber, fish, vegetable protein and vegetable fat from olive oil, high consumption of nuts, low consumption of sweetened/carbonated beverages, refined grain products, and partially hydrogenated or trans fat) are inversely associated with all adiposity indexes [15]. Consumption frequencies and suggested portion sizes are represented within a pyramid. At the base of the pyramid are the foods that should be eaten at each meal, along with the Mediterranean diet’s social, cultural, and environmental aspects. Among these foods are fruits and vegetables consumed 1–2 times a day and grains consumed in 2–3 servings of bread, pasta, couscous, rice, oats, or other grains. In the middle of the pyramid are foods that should be consumed daily, including 2 servings of milk and yoghurt, 1 serving of breakfast products such as cookies and cereals, 2–3 servings of extra virgin olive oil as the main source of fat, and finally, herbs and spices for flavoring. At a higher level are foods that should be consumed weekly, such as animal products, which should be alternated with legumes and combined with cereal consumption. Potatoes should also be consumed weekly. At the pyramid’s apex are foods such as sweets that should be eaten only once weekly [14]. By following this diet, one can observe a decrease in BMI, fat mass, and lean mass, as well as reductions in glucose levels, total cholesterol, TGC, and LDL-c, along with an increase in HDL-c and a reduction in the presence of metabolic syndrome components. A 2014 study involving obese pediatric subjects reported a reduction in BMI that was not accompanied by weight loss but with an increase in height after 4 months of the Mediterranean diet [16]. The Mediterranean diet has long been defined as being protective against various non-communicable diseases. It is considered a model of a healthy diet, particularly after the publication of the first results of the PREDIMED study, which demonstrated a reduction in mortality from cardiovascular disease in subjects adhering to this diet [17]. Due to its richness in antioxidants and other bioactive molecules, the Mediterranean diet plays an important protective role against atherosclerosis by improving blood lipid levels, oxidative and inflammatory status, and gene expression associated with the development of cardiovascular disease [18]. The beneficial effect of the Mediterranean diet on hepatic lipid metabolism is mainly due to the consumption of fatty acids, particularly MUFAs, and a proper ratio of omega-6 to omega-3, facilitated by the abundant consumption of vegetables, legumes, nuts, olive oil, and fish [19]. MUFAs can reduce fat deposition in the liver and promote fat deposition in adipose tissue. The lipid profile improves due to reduced TGC and low-density lipoprotein (VLDL) levels without reducing HDL-c [20]. The Mediterranean diet is also characterized by a low intake of saturated fat and a high consumption of water-soluble fiber, which leads to the sequestration of cholesterol and slows down its absorption [19,21]. Other components of the Mediterranean diet are polyphenols, which can reduce TGC levels, exhibit important antioxidant effects, and have cholesterol-lowering and anti-inflammatory properties. Studies have shown how the Mediterranean diet improves LDL-C characteristics, making it particularly atherogenic [22]. Physical activity represents a further modifiable risk factor for obesity. Children with DS usually do not reach the recommended 60 min of moderate to vigorous physical activity three times a week [23]. The involvement in physical activities depends on clinical problems linked to DS, such as reduced peak aerobic capacity (VO2_peak_) [24] and reduced muscular strength [25], in addition to an altered functional profile, such as lower cognitive and behavioral skills [26]. VO2_peak_ in DS youth and adults is reduced compared with the general population and is associated with the ability to perform lower-intensity activities and a faster time to exhaustion of strength [8]. The causes of reduced VO2_peak_ depend on autonomic dysfunction, which is associated with a similarly reduced peak heart rate (HR) due to an enlarged tongue typical of the syndrome; this may limit ventilation for high-intensity efforts and metabolic alterations. Their participation may be adversely affected by other factors, such as craniofacial malformations that may hinder speech and hearing development; the nearly 2-year delay in communication skills; hypotonia; excessive ligamentous laxity, with the development of abnormalities of alignment and movement of the foot, ankle, knee, and hip; the inability of sensory integration; memory deficits; and difficulties in problem solving and planning. Although there are no specific guidelines for physical activity in children with DS, recommendations for individuals with cognitive disabilities include various recreational activities during the day [8]. Children with DS are typically less sedentary due to higher rates of psychiatric and behavioral disorders [27], but they engage more in low-intensity activities compared to the general pediatric population [26]. The functional component in children with DS includes reduced cognitive, behavioral, and motor aspects, with a tendency toward slower, safer, and more accurate movement patterns due to interactions between neurological, environmental, and task-associated limitations. Physical inactivity in children with DS may also result from tendencies for exclusion from hands-on activities; lack of accessibility, inclusiveness, and appropriate programs; negative behaviors; transportation difficulties; family upbringing; and lack of friends [26]. Dyslipidemia is strictly related to childhood obesity [28]. Children with DS often develop an unfavorable lipid profile, characterized by higher LDL-cholesterol (LDL-c) and lower HDL-cholesterol (HDL-c) levels [29]. Previous evidence demonstrated that 58% of children with DS present dyslipidemia, and the most common alteration is a reduction in HDL-c (45.9% of cases), followed by hypertriglyceridemia (26.2%). Garcia-de la Puente et al. demonstrated a strong association between obesity and dyslipidemia, particularly between BMI and triglyceride levels [30]. The prevalence of dyslipidemia is much higher than in the healthy pediatric population, where dyslipidemia varies between 15 and 30% [31]. Further studies demonstrated that children with DS may have a worse lipid profile than their siblings, regardless of weight status [29]. Starting from this evidence, it is possible to speculate that modifying lipid levels is possible through interventions in dietary patterns and physical activity [22,32,33]. Children with DS may profit from a multidisciplinary approach in which the pediatrician pays special attention to their nutritional status. Starting with this clinical complexity, children with DS require a complex care system with a multispecialist approach. For this reason, specific and multidisciplinary centers have developed over time worldwide. The example is represented by the Netherlands, where specific pediatric clinics are dedicated to DS, which provide appointments, including visits with the pediatrician, physiotherapist, otolaryngologist, and other specialists, all on the same day. These clinics are called “Down teams” [34]. In other countries, the management of DS seems relatively decentralized, with few centers providing multidisciplinary care. Healthcare centers follow most individuals with DS, but it is possible that many of the recommended treatments and routine examinations for this population are not carried out promptly and comprehensively. This may be due to a failure to perform tests indicated by the pediatrician, a lack of request by the physician for certain examinations to be conducted because they are unaware of the specific issues associated with DS, or a failure to communicate test results to parents [35]. The complexity of these treatments concerns the medical field and other aspects that influence the quality of life of the child with DS and the family. Given the clinical complexity of children with DS, a holistic approach must be emphasized, considering health problems and other aspects, such as personality, lifestyle, and physical and social development [34]. Moreover, nutritional and physical activity-related issues should not be underestimated as part of the multispecialty evaluation of the child with DS. It is fundamental that pediatricians know this aspect. The importance of a specialized outpatient clinic that follows these patients over time in a dedicated manner and with a multidisciplinary approach is emphasized. In the literature, outpatient experiences of individuals with DS are reported, but not much information regarding the nutritional plan and behavioral advice on physical activity is present. We designed a prospective study to evaluate the impact of dietary and physical advice on children with DS, investigating changes over time in anthropometric data, laboratory data, adherence to the Mediterranean diet, and physical activity following periodic check-ups.

## 2. Materials and Methods

This is a prospective study evaluating the impact of dietary and physical advice on children with DS. Forty-four children with DS aged between 2 and 17 years were enrolled in the study. The participants were recruited between December 2020 and May 2021 during the outpatient visits for DS at the Mother and Child Hospital in Verona, Italy. Children with active diseases, such as orthopedic or feeding difficulties, neoplasms, and infectious diseases, were not eligible for enrollment. The parents or legal guardians signed the informed consent. This study was approved by the Ethics Commission for Clinical Trials of the Provinces of Verona and Rovigo (DONUT STUDY, code number 66508, 7 December 2020; amendment, code number 45686, 25 July 2022). Participants and their parents received specific behavioral instructions from a specialized pediatrician about the benefits of adherence to the Mediterranean diet and the importance of engaging in physical activity. Participants followed a regular follow-up program, undergoing clinical, anthropometric, and laboratory evaluations involving lipidic profiles every six months between December 2020 and May 2023. Adherence to the Mediterranean diet was assessed using the Mediterranean diet “KIDMED—Mediterranean Diet Quality Index" in children and adolescents [36]. Physical activity was assessed using the “Godin–Shepard Leisure-Time Physical Activity Questionnaire” [37]. The outcome measures included the Godin score (physical activity), KIDMED score (adherence to the Mediterranean diet), BMI z-score, and lipid profile (HDL, LDL, and TGC), which were compared between baseline visits and 2-year follow-up visits. Statistical analyses were performed with R 4.3 (R Foundation for Statistical Computing, Vienna, Austria) [38]. Continuous variables were summarized as the mean and standard deviation (SD) and were compared between the baseline visit and 2-year follow-up visit using the Student’s *t*-test for paired data. The change over time was compared between sex and age groups using Student’s *t*-test. Categorical variables were summarized as numbers and percentages and compared between baseline and 2-year follow-up visits using the McNemar test. All tests were two-tailed, and a *p*-value of less than 0.05 was considered statistically significant.

## 3. Results

The study included 44 children with DS (24 males and 20 females; mean age 9 years, SD 3). The most prevalent comorbidities were heart diseases (31/44, 70%), ocular pathologies (29/44, 65%), and otolaryngological pathologies (22/44, 50%). Table 1 summarizes the adherence to the Mediterranean diet and physical activity at baseline and 2-year follow-up visits.

During the study period, the mean Godin score increased from 17 to 22 points (mean difference 5 points, 95% CI 3 to 8; *p* = 0.004), with an improvement in the level of physical activity (*p* = 0.002) (Table 1). The mean KIDMED score was 6 points both in 2020 and 2023 (mean difference 0 points, 95% CI −1 to 0; *p* = 0.32), without a significant change in diet adherence (*p* = 0.50) (Table 1). Figure 1 displays the distribution of the Godin score and KIDMED score at the baseline visit and 2-year follow-up visit. The change in Godin score was not different according to sex (*p* = 0.33) or prepubertal/pubertal ages (*p* = 0.85). Similarly, the change in the KIDMED score was not different according to sex (*p* = 0.88) or prepubertal/pubertal ages (*p* = 0.88).

During the study period, there was a decrease in the BMI z-score (mean difference: −0.4, 95% CI −0.7 to −0.1; *p* = 0.006), which was coupled with less obese children in the follow-up visit (Table 2). HDL decreased (mean difference: −3 mg/dl, 95% CI −6 to −1; *p* = 0.03) and LDL increased (mean difference: 5 mg/dl, 95% CI 0.1 to 10; *p* = 0.04), while TGC and dyslipidemia levels did not significantly change over time (Table 2). Figure 2 displays the distribution of the BMI and lipid profile at baseline and 2-year follow-up visits. To provide more detailed information to the reader, the trend of the lipid profile for each subject is shown in Appendix A. The change in LDL was different between prepubertal and pubertal subjects (*p* = 0.01), while age was not associated with different changes in the BMI z-score (*p* = 0.85), HDL (*p* = 0.31), or TGC (*p* = 0.49). Moreover, sex was not associated with different changes in the BMI z-score (*p* = 0.49), LDL (*p* = 0.15), HDL (*p* = 0.13), or TGC (*p* = 0.87). 

## 4. Discussion

Our findings showed that behavioral instructions regarding nutrition and physical activity were associated with some changes in lifestyle habits and nutritional status among children with DS. These subjects are prone to developing inappropriate food and physical habits that affect their health [8,26]. Hence, we hypothesized that they might benefit from regular dietary and physical advice from specialized pediatricians. In our participants, counseling was associated with increased physical activity as measured by the Godin questionnaire. This translated into more children engaging in an active lifestyle compared to those being moderately or insufficiently active. Interestingly, previous studies on the DS population showed that physical activity tended to decrease with age, especially in those engaging in high-intensity physical activity at initial assessments [23,39]. Thus, our results suggest that appropriate and regular advice on physical activity may counteract this tendency and benefit children with DS. On the other hand, counseling did not improve adherence to the Mediterranean diet as measured using the KIDMED questionnaire. Overall, most participants reported a moderate/high adherence to the Mediterranean diet, and only a few changes occurred during the 2-year study period. While maintaining such adherence over time provided an encouraging message, we believe further efforts are needed to achieve a higher adherence rate to the Mediterranean diet. The lack of a significant increase in adherence to the Mediterranean diet, as indicated by the KIDMED score, could potentially be attributed to several factors. One significant challenge is that children with DS often consume foods in liquid form due to difficulties with cutlery use [9]. The Mediterranean diet, rich in whole foods like fruits, vegetables, grains, and lean proteins, may not align well with their dietary preferences or practical constraints. A few Mediterranean options can be consumed in liquid form, such as olive oil, milk, and yogurt, while most are solid foods [14]. This mismatch may hinder effective adherence to the Mediterranean diet for children with DS, despite efforts to promote it. The limited availability of liquid-form Mediterranean foods might be a key reason for the observed lack of improvement in adherence. Tailored dietary interventions addressing the unique needs of children with DS may be necessary for effective dietary management. Improved dietary habits, with increased adherence to the Mediterranean diet, aim to improve children’s health status, particularly regarding BMI and glucose and lipid profile [16]. We recorded a reduction in the BMI z-score, translating into less obese children at the 2-year follow-up visit. The youth DS population is prone to overweight and obesity, with consequences on their health status [5]. In our study, behavioral instructions regarding nutrition and physical activity offered to children and parents during regular outpatient visits seemed to have positively influenced the nutritional status. At the same time, the improvement in physical activity and the maintenance of moderate/high adherence to the Mediterranean diet have not accompanied some changes in the lipid profile. Although it is generally known that diet and physical activity interventions may improve the lipid profile in children of pediatric age [33], there are no prospective studies in children with DS. A cross-sectional study by Adelekan et al. found that non-obese children with DS had an unfavorable lipid profile compared to their siblings, regardless of body weight [29]. In addition, de la Piedra et al. did not find any association between dyslipidemia and nutritional status in children with DS [31]. Of note, overall dyslipidemia did not change over time in our sample. We do not have a clear explanation for the conflicting findings on physical activity, obesity, and dyslipidemia. We may speculate that the pediatric DS population may be unique and different from the general population regarding those aspects, which requires further investigations with a large sample size. In our cohort, the prevalence of dyslipidemia was lower compared to other studies in children with DS (30–34% vs. up to 60%) [29,30,31]. A previous study suggested that inadequate nutritional education in parents led to inappropriate diet management [12]. Therefore, the education of parents or guardians of children with DS should focus on adopting a more appropriate diet and physical activity, preventing obesity and other comorbidities such as dyslipidemia. Despite this, affected individuals are often well-nourished, but to improve the child’s health, it is considered essential to educate the parents or guardians of the children to follow a correct diet [12]. The environment within a child’s family home plays a crucial role in shaping their lifelong behaviors. It is fundamental in establishing habits that endure into adulthood. Parents should be provided with information and guidance on how and what to feed their children [40]. The benefit of parental nutrition education in improving children’s health outcomes has been demonstrated [41]. Parents seem to have a high degree of control in modeling their children’s eating behaviors [42]. Many children who attend childcare facilities have eating habits shaped by parents and childcare providers [43]. In this context, collaboration between parents and providers is important to ensure optimal nutrition for the children under their care. Effective communication between childcare providers and parents is essential, and this has been recognized by Head Start (HS), the primary US funder of childcare services for low-income families, and by the Academy of Nutrition and Dietetics (Academy), the largest organization of nutrition professionals. The Head Start Performance Standards state that staff and parents must collaborate to identify the nutritional needs of each child. A lack of coordination or effective communication between parents and providers was reported. Specifically, center directors reported a lack of parental involvement, center staff reported limited time, and family care providers reported a lack of healthy eating at home as an obstacle [44]. A study identified five barriers or challenges that providers encountered when communicating with parents about their child’s nutrition. Firstly, parents are often perceived as too busy during drop-off or pickup times, leaving little opportunity for discussion. Secondly, providers express concerns about some parents offering unhealthy foods to children due to convenience. Additionally, while parents may engage in discussions about food issues, they are less receptive to conversations about nutrition. Providers also worry about offending parents when discussing nutrition and question whether parents will be receptive and interested in the information provided. To overcome these barriers, childcare providers have outlined strategies for better communication with parents regarding their children’s nutrition, focusing on five key themes. These strategies aim to promote children’s well-being through parental education on nutrition. Providers recognize the benefits of discussing nutrition with parents to improve the nutritional environment at home, prevent health issues like obesity, and enhance overall child health. Providers view their relationship with parents as a collaborative partnership, where both parties work together to promote children’s health through education. Providers use federal and state regulations to guide discussions on nutrition, minimizing conflicts and fostering positive communication. Providers simplify communication with parents regarding nutrition-related matters by integrating nutritional policies into their center-level operations. They prioritize mutual respect to overcome communication barriers, emphasizing the importance of respectful interactions for effective discussions on children’s health [45]. Strategies intended to improve communication between parents and caregivers and educate parents about child nutrition could also be applied to other areas, such as promoting physical activity. Nutritional and physical activity guidelines can be applied to the general and pediatric populations with DS. Despite potential challenges, individuals with DS can adopt a proper diet and approach physical activity similarly to children without DS [46]. Hence, these strategies also benefit children’s health and well-being for individuals with DS. Regarding the potential benefits resulting from the increase in physical activity and decrease in BMI z-score, it is conceivable that lifestyle changes could contribute to an overall improvement in well-being among children with DS and their families. We may speculate that the improvements in physical activity levels and the maintenance of good adherence to the Mediterranean diet observed in our pediatric population with DS are partially due to effective educational intervention in the outpatient setting. Our study has some limitations. First, the small sample size suggests caution in interpreting the findings. Second, the questionnaires on diet and physical activity were filled in by parents or legal guardians of children with DS; hence, the recorded information may be prone to subjective bias. Third, the change in the relevant outcome measures was assessed over 2 years, and further studies are required to investigate the possible benefits of the intervention in the long term. Fourth, the study was conducted in a single center in northern Italy; hence, the generalizability of the findings may be limited to similar settings.

## 5. Conclusions

Our study suggested that regular counseling on diet and physical activity regarding overweight and obesity may benefit children with DS. Children with DS can improve their health status in terms of a reduction in the BMI z-score and an increase in the level of physical activity. This goal can be best achieved when a specific follow-up is dedicated to these children, with the potential effect of keeping under control the unavoidable worsening of dyslipidemia. Specialized outpatient clinics for children with DS may consider implementing such counseling in association with monitoring growth and development, recognizing risk factors for the development of obesity in this special population at pediatric age, and implementing awareness strategies to improve health. Larger long-term investigations are needed to confirm these findings. In addition, further studies should identify more appropriate interventions to increase adherence to the Mediterranean diet and explore the association between counseling and changes in the lipid profile of children with DS. An intervention may address the preference for liquid foods in children with DS, attributed to cutlery difficulties. The emphasis on whole foods in the Mediterranean diet further exacerbates this challenge, as there are limited options in liquid form, potentially hindering adherence. A worsening lipid profile is known to increase cardiovascular mortality, so the long-term monitoring of these modifiable cardiovascular risk factors seems to be crucial for reducing the risk of comorbidities in adults. Establishing outpatient clinics for DS in adulthood may improve the level of care and health of such patients. Educating parents about children’s nutrition is crucial for establishing lifelong healthy eating habits. Effective communication between parents and childcare service providers is essential for promoting children’s well-being. The strategies outlined by the operators aim to enhance this communication and empower parents through education. These efforts benefit the general pediatric population and individuals with DS, highlighting the importance of collaborations in promoting children’s health and nutrition.

## Figures and Tables

**Figure 1 diseases-12-00068-f001:**
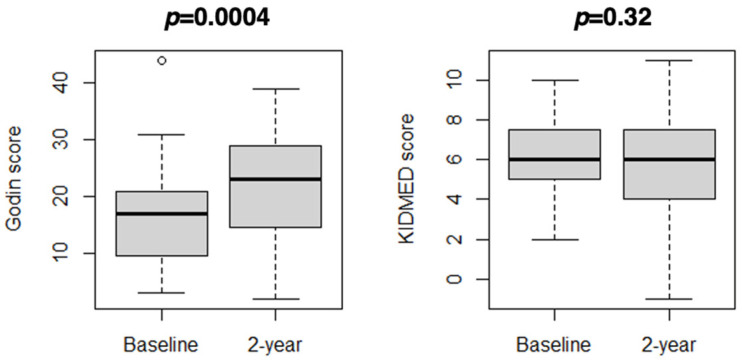
Distribution of Godin score (physical activity) and KIDMED score (adherence to the Mediterranean diet) at the baseline visit and 2-year follow-up visit: box and whiskers plots.

**Figure 2 diseases-12-00068-f002:**
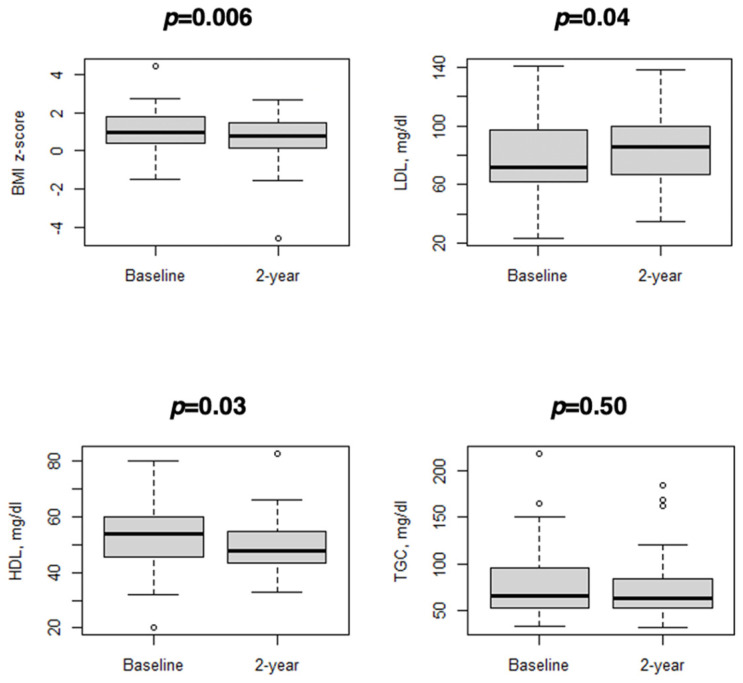
Variation in BMI z-score and lipid profile at baseline visit and 2-year follow-up visit: box and whiskers plots.

**Table 1 diseases-12-00068-t001:** Comparison of physical activity (Godin) and adherence to the Mediterranean diet (KIDMED) at the baseline visit and 2-year follow-up visit.

Variable	Baseline (n = 44)	2-Year Follow-Up (n = 44)	*p*-Value
Godin score:			0.0004
Mean (SD)	17 (8)	22 (10)	
Median (IQR)	17 (10–21)	23 (15–29)	
Min; max	3; 44	2; 39	
Physical activity, n (%):			0.002
Active	6 (14%)	22 (50%)	
Moderately active	22 (50%)	12 (27%)	
Insufficiently active	16 (36%)	10 (23%)	
KIDMED score:			0.32
Mean (SD)	6 (2)	6 (3)	
Median (IQR)	6 (5–7)	6 (4–7)	
Min; max	2; 10	1; 11	
Diet adherence, n (%):			0.50
High	11 (25%)	12 (27%)	
Moderate	28 (64%)	22 (50%)	
Low	5 (11%)	10 (23%)	

**Table 2 diseases-12-00068-t002:** Comparison of the BMI z-score and lipid profile at the baseline visit and 2-year follow-up visit.

Variable	Baseline (n = 44)	2-Year Follow-Up (n = 44)	*p*-Value
BMI z-score:			0.006
Mean (SD)	1.1 (1.2)	0.7 (1.2)	
Median (IQR)	1.0 (0.4–1.8)	0.8 (0.1–1.4)	
Min; max	−1.5; 4.5	−4.6; 2.7	
BMI category, n (%):			0.05
Normal weight	27 (61%)	29 (66%)	
Overweight	11 (25%)	13 (30%)	
Obese	6 (14%)	2 (4%)	
LDL, mg/dl:			0.04
Mean (SD)	78 (27)	83 (24)	
Median (IQR)	72 (62–97)	86 (67–100)	
Min; max	23; 141	35; 138	
HDL, mg/dl:			0.03
Mean (SD)	53 (12)	50 (10)	
Median (IQR)	54 (45–60)	48 (43–55)	
Min; max	20; 80	33; 83	
TGC, mg/dl:			0.50
Mean (SD)	79 (39)	73 (33)	
Median (IQR)	66 (53–96)	63 (52–84)	
Min; max	33; 219	31; 184	
Dyslipidemia, n (%):	13 (30%)	15 (34%)	0.77

## Data Availability

Data are contained within the article and Appendix A.

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
