# Peer review of "Lipid Profile, Eating Habit, and Physical Activity in Children with Down Syndrome: A Prospective Study"

_diseases, 2024, doi:10.3390/diseases12040068_

Round 1

Reviewer 1 Report

Comments and Suggestions for Authors

The authors proposed that providing regular guidance on diet and physical activity, particularly in relation to physical activity and obesity, could potentially improve the health outcomes of children with down syndrome. Children with down syndrome have the opportunity to enhance their health by lowering their BMI z-score and enhancing their level of physical activity. The authors conducted a well-developed prospective study; however, there are some minor details that need improvement:

- In lines 45-47, the authors state, "These factors contribute to lower energy consumption and thus a requirement of 500-800 calories per day less than in the unaffected population; this increases the risk of excessive calorie intake and consequent weight gain." However, it seems necessary to reference the bibliographic citation from which these data were extracted.

- In the manuscript, the authors focus on the effects of the Mediterranean diet on children with Down syndrome. Is there another similar study in the literature focusing on a different type of diet? If so, it would be interesting to compare the results obtained.

- In this article, the authors included 24 males and 20 females, but they studied all individuals together. Have they observed differences in physical activity (Godin) or adherence to the Mediterranean diet (KIDMED) when comparing males to females? The same in BMI z-score and lipid profile.

- Similarly, the mean age studied is 9 years. Have the authors observed differences in physical activity (Godin) or adherence to the Mediterranean diet (KIDMED) between prepubertal and pubertal stages? The same in BMI z-score and lipid profile.

-The authors, in the results obtained regarding adherence to the Mediterranean diet (KIDMED score), do not observe a significant increase in it. In the introduction, the authors explain that children with DS have difficulties in the use of cutlery, which promotes the consumption of food in liquid form (line 60). However, within the Mediterranean diet, there are very few foods that can be consumed in liquid form (olive oil, milk, and yogurt; the vast majority are solid products). Do the authors believe that this could be one of the reasons why an increase in the KIDMED score was not obtained?

- Throughout the manuscript, the authors highlight the crucial role that the family plays in the lives of children with DS. What reasons have the families of the 44 children included in the study given for not adhering to the Mediterranean diet? Additionally, have the authors observed any additional benefits resulting from the increase in physical activity and decrease in BMI z-score?

Comments on the Quality of English Language

Minor editing of English language required

Author Response

Reviewer #1:

The authors proposed that providing regular guidance on diet and physical activity, particularly in relation to physical activity and obesity, could potentially improve the health outcomes of children with down syndrome. Children with down syndrome have the opportunity to enhance their health by lowering their BMI z-score and enhancing their level of physical activity. The authors conducted a well-developed prospective study; however, there are some minor details that need improvement:

In lines 45-47, the authors state, "These factors contribute to lower energy consumption and thus a requirement of 500-800 calories per day less than in the unaffected population; this increases the risk of excessive calorie intake and consequent weight gain." However, it seems necessary to reference the bibliographic citation from which these data were extracted.

We have added this bibliographic reference.

In the manuscript, the authors focus on the effects of the Mediterranean diet on children with Down syndrome. Is there another similar study in the literature focusing on a different type of diet? If so, it would be interesting to compare the results obtained.

To the best of our knowledge, there are no studies that deepened a specific type of diet in children affected by Down Syndrome. Bialek Drawta et al. (Białek-Dratwa A, Żur S, Wilemska-Kucharzewska K, Szczepańska E, Kowalski O. Nutrition as Prevention of Diet-Related Diseases-A Cross-Sectional Study among Children and Young Adults with Down Syndrome. Children (Basel). 2022 Dec 24;10(1):36. doi: 10.3390/children10010036. PMID: 36670587; PMCID: PMC9856910.) developed a cross-sectional study, demonstrating that most subjects, including adults and children with Down Syndrome, are usually fed normally, but the parents make nutritional mistakes.

In this article, the authors included 24 males and 20 females, but they studied all individuals together. Have they observed differences in physical activity (Godin) or adherence to the Mediterranean diet (KIDMED) when comparing males to females? The same in BMI z-score and lipid profile.

We thank the Reviewer for the comment, which enriched our manuscript. We added such information on pages 5-6 in the revised version. We found no differences between males and females in the change over time of the study outcomes (physical activity, adherence to the Mediterranean diet, BMI z-score, lipid profile).

Similarly, the mean age studied is 9 years. Have the authors observed differences in physical activity (Godin) or adherence to the Mediterranean diet (KIDMED) between prepubertal and pubertal stages? The same in BMI z-score and lipid profile.

We added such information in the new version of the manuscript on pages 5-6. We found that LDL change over time was different between prepubertal and pubertal subjects (p=0.01), while there were no significant differences in the other study outcomes (physical activity, adherence to the Mediterranean diet, BMI z-score, HDL, TGC).

The authors, in the results obtained regarding adherence to the Mediterranean diet (KIDMED score), do not observe a significant increase in it. In the introduction, the authors explain that children with DS have difficulties in the use of cutlery, which promotes the consumption of food in liquid form (line 60). However, within the Mediterranean diet, there are very few foods that can be consumed in liquid form (olive oil, milk, and yogurt; the vast majority are solid products). Do the authors believe that this could be one of the reasons why an increase in the KIDMED score was not obtained?

We thank the Reviewer for this feedback. We should consider your suggestion into our analysis and future research. The lack of a significant increase in adherence to the Mediterranean diet, as indicated by the KIDMED score, could potentially be attributed to several factors, that we have deepened in the main text.

Throughout the manuscript, the authors highlight the crucial role that the family plays in the lives of children with DS. What reasons have the families of the 44 children included in the study given for not adhering to the Mediterranean diet? Additionally, have the authors observed any additional benefits resulting from the increase in physical activity and decrease in BMI z-score?

The questionnaire utilized in our study did not specifically inquire why families were not adhering to the Mediterranean diet. However, this aspect offers an intriguing opportunity for future research to explore the factors influencing dietary decisions among families of children with DS. Regarding the potential benefits resulting from the increase in physical activity and decrease in BMI z-score, our study primarily focused on evaluating adherence to the Mediterranean diet and its impact on metabolic health indicators. While we did not extensively investigate secondary outcomes such as quality of life for patients and families, it is conceivable that these lifestyle changes could contribute to an overall improvement in well-being.

Reviewer 2 Report

Comments and Suggestions for Authors

The manuscript by Pecoraro L. et al. is a prospective study evaluating the adherence to the mediterranean diet and physical activity in children with Down Syndrome.

The manuscript is well written and it provides a hint to health status improvement in DS pediatric population, however some concerns need to be addressed:

-Despite the reduction in BMI and the increase in the Godin score, lipid profile get worse after 2 years. To better understand this unexpected result, values should be expressed as median (SD, min and max) in Table 2 and an additional figure with a before-after plot showing the trend for each subject should be provided, as it could help to understand why lipid profile get worse and if it is caused by few outliers.

-Dyslipidemia is associated with vitamin D deficiency and can contribute towards cardiovascular diseases even in normal weight and under weight subjects. Population studies show that people with lower vitamin D levels are more likely to have high cholesterol, although this doesn't prove a “cause and effect” relationship. What about the vitamin D status in your samples, and in DS children compared to healthy children? Could a vitamin D supplementation, coupled to Mediterranean diet and physical activity, be beneficial on DS children lipid profile?

-Introduction is very long, I suggest to reorganize it

Comments on the Quality of English Language

Nothing to comment

Author Response

Reviewer #2:

The manuscript by Pecoraro L. et al. is a prospective study evaluating the adherence to the mediterranean diet and physical activity in children with Down Syndrome.

The manuscript is well written and it provides a hint to health status improvement in DS pediatric population, however some concerns need to be addressed:

Despite the reduction in BMI and the increase in the Godin score, lipid profile get worse after 2 years. To better understand this unexpected result, values should be expressed as median (SD, min and max) in Table 2 and an additional figure with a before-after plot showing the trend for each subject should be provided, as it could help to understand why lipid profile get worse and if it is caused by few outliers.

We thank the Reviewer for the suggestion, which improved the clarity for the reader. The new version of the manuscript provided more summary indicators in tab.2 (median, IQR, mean, SD, min, max). The same indicators were also added in tab.2. Moreover, we added supplementary figures 1-3 in the new version of the manuscript displaying the before-after trend of the lipid profile for each subject as requested.

Dyslipidemia is associated with vitamin D deficiency and can contribute towards cardiovascular diseases even in normal weight and under weight subjects. Population studies show that people with lower vitamin D levels are more likely to have high cholesterol, although this doesn't prove a “cause and effect” relationship. What about the vitamin D status in your samples, and in DS children compared to healthy children? Could a vitamin D supplementation, coupled to Mediterranean diet and physical activity, be beneficial on DS children lipid profile?

We thank the Reviewer for the insightful comment. Vitamin D plasmatic levels have not been deepened in our sample. Given the known association between dyslipidemia and vitamin D deficiency and its potential impact on cardiovascular health, the suggestion of exploring the potential benefits of vitamin D supplementation alongside the Mediterranean diet and physical activity is intriguing and warrants further investigation.

Introduction is very long, I suggest to reorganize it

The introduction was structured as follows:

  1. Introduction to Down syndrome
  2. Phenotypic variability and comorbidities of DS
  3. Challenges in managing weight in DS
  4. Dietary habits and nutritional challenges in DS
  5. Mediterranean diet as a dietary model
  6. Effects of the Mediterranean diet on health parameters
  7. Role of physical activity in managing obesity and health in DS
  8. Dyslipidemia and Cardiovascular Risk in DS
  9. Multidisciplinary approach to DS management
  10. The need for specialized healthcare centers for DS
  11. Challenges in comprehensive care for DS individuals
  12. Importance of nutritional and physical activity counselling
  13. Design of prospective study for DS management evaluation

Regarding the length, the editor requested that 4000 words be developed for the article, and the introduction should be increased. We would like not to modify it.